Exploring women’s oxytocin responses to interactions with their pet cats

Johnson Elizabeth A. Elizabeth.johnson@unlv.edu
Portillo Arianna
Bennett Nikki E.
Gray Peter B.
Department of Anthropology, University of Nevada, Las Vegas , Las Vegas , NV , United States of America
Vonk Jennifer
Electronic publication date: 2021 Nov 12
Publication date: 2021
Volume: 9
Electronic Location ID: e12393
Received 2021 Jun 15; Accepted 2021 Oct 5
Copyright: ©2021 Johnson et al.
Copyright year: 2021
Copyright holder: Johnson et al.
License: This is an open access article distributed under the terms of the Creative Commons Attribution License, which permits unrestricted use, distribution, reproduction and adaptation in any medium and for any purpose provided that it is properly attributed. For attribution, the original author(s), title, publication source (PeerJ) and either DOI or URL of the article must be cited.
License URL: https://creativecommons.org/licenses/by/4.0/

Keywords: Anthrozoology, Human–animal interactions, Oxytocin, Domestic cats Felis catus, Animal Behavior

Funding: University of Nevada, Las Vegas (UNLV) Department of Anthropology UNLV Graduate & Professional Student Association UNLV University Libraries Open Article Fund This work was supported by the University of Nevada, Las Vegas (UNLV) Department of Anthropology, and the UNLV Graduate & Professional Student Association. The publication fees for this article were supported by the UNLV University Libraries Open Article Fund. The funders had no role in study design, data collection and analysis, decision to publish, or preparation of the manuscript.

==============================
Background

Extensive research has evaluated the involvement of the neuropeptide oxytocin (OT) in human social behaviors, including parent-infant relationships. Studies have investigated OT’s connection to human attachment to nonhuman animals, with the majority of the literature focusing on domestic dogs (Canis lupis familiaris). Utilizing what is known about OT and its role in maternal-infant and human-dog bonding, we apply these frameworks to the study of human-domestic cat (Felis catus) interactions.

Methods

We investigated changes in salivary OT levels in 30 U.S. women of reproductive age before and after two conditions: reading a book (control) and interacting with their pet cat. Participant and cat behavioral patterns during the cat interaction condition were also quantified to determine if differences in women’s OT concentrations were associated with specific human and cat behaviors.

Results

Our results revealed no changes in women’s OT levels during the cat interaction, relative to the control condition, and pre-cat interaction OT levels. However, differences in women’s OT concentrations were correlated with some human-cat interactions (e.g., positively with petting cat and cat approach initiation, negatively with cat agonistic behavior) but not all observed behaviors (e.g., use of gentle or baby voice) coded during human-cat interactions.

Discussion

This study is the first to explore women’s OT in response to interactions with their pet cat and has identified distinct human and cat behaviors that influence OT release in humans.

Introduction

As a social species, kinship and the ability to form social attachments have been central to human evolution (Feldman, 2017). Contributing to this evolutionary success has been the selection for neurophysiological processes that establish and maintain cooperative relationships (Carter, 2017). Oxytocin (OT) is a neuropeptide hormone involved in a variety of mammalian behaviors to include sociosexual behaviors (e.g., Witt, Winslow & Insel, 1992) and parental care (e.g., Insel & Young, 2001). The OT system is potentiated by estrogen and exhibits some sex-specific effects (Carter, 2017), though the underlying neurological processes exist across both sexes and impact both male and female social cognition (Anacker & Beery, 2013; Insel & Young, 2001).

Studies evaluating human OT levels following parent-infant interactions show increased peripheral OT levels for mothers after affectionate behaviors (e.g., kissing) and for fathers following stimulatory interactions (e.g., object presentation; Scatliffe et al., 2019; Gettler et al., 2021). Nonhuman animal models have highlighted OT’s influence on mother-infant bonding. When given exogenous OT, female rats demonstrate more maternal behaviors (e.g., pup grooming; Insel, 1997). Rats who are unable to uptake OT (e.g., brain lesions) actively avoid their newborn pups (Van Leengoed, Kerker & Swanson, 1987). Though less clear due to the ethical constraints of human research, research into human maternal behavior suggests OT is related to affectionate touch, mutual eye gazing, and positive mood (Bell, Erickson & Carter, 2014).

While intraspecific social behaviors are observed extensively in mammals, humans are also known to have affective relationships with allospecifics, or members of other species, that can be understood in terms of kinship (Charles, 2014). Most notable is the human-domestic dog (Canis lupus familiar) relationship (Nagasawa, Mogi & Kikusui, 2009). Domestic dogs possess social cognitive functions analogous to human infants (Buttner, 2016) and have been observed using social referencing cues similar to that of mother-infant interactions (e.g., Téglás et al., 2012). Hare & Tomasello (2005) suggest the domestic dog’s ability to read human communicative behaviors such as pointing gestures result from human-dog coevolution. Even further, these abilities are reflected in neurobiological processes that interact with OT to mediate these human-dog social interactions (see Buttner, 2016; Powell et al., 2019 for complete reviews).

Considering what is known about OT’s involvement in human social behaviors and the sociocognitive abilities of domestic dogs, extensive research has explored OT’s role in human-canine interactions (Buttner, 2016). In evaluating the possible physiological processes involved in human-dog interactions, Odendaal (2000) measured human plasma OT levels before and after the following conditions: quiet book reading, interacting with an unfamiliar dog, and participants interacting with their pet dogs. For both familiar and unfamiliar dog interactions, OT concentrations increased with larger increases observed for familiar dog interactions (Odendaal, 2000). In looking at specific behaviors occurring during human-dog interactions, Nagasawa and colleagues (2015) investigated the effects of mutual eye gazing between humans and familiar dogs. Their results showed increased peripheral OT levels, measured in human urine, were associated with higher levels of mutual gaze.

Domestic cats (Felis catus) have been domesticated for thousands of years, and are now one of the most common household pets (Hu et al., 2014). They have garnered attention for their social cognitive abilities and attachments with humans (Vitale, Behnke & Udell, 2019), though research has primarily focused on human-dog relationships. Just as domestic dogs use social referencing, Merola et al. (2015) found that cats also reference their owners when presented with an unfamiliar object. More recent research has also shown the similarities in dog and cat use of human cues (see Chijiiwa et al., 2021). Other studies have suggested cats may positively influence health, with heart rate and blood pressure decreasing following the cat interaction and this decrease being more pronounced in cat owners (Dinis & Martins, 2016). Although the human-cat relationship has been highlighted more recently, only one published study has evaluated domestic cats as a stimulus for human OT response. Curry et al. (2015) analyzed plasma OT concentrations from human samples collected before and after lab-based interactions with unfamiliar dogs and unfamiliar cats. Human OT levels tended to decrease during the cat interaction, though less so if the humans had a history of keeping pet cats. In addition to OT levels being influenced by the participant’s previous experiences with companion animals, the participant’s background with the animals (i.e., familiar versus unfamiliar) has also been found to be an important variable to consider when designing human-animal OT studies. As such, other studies have demonstrated decreased OT responses in human participants interacting with unfamiliar dogs (Handlin et al., 2012) or viewing photos of unfamiliar dogs (Powell et al., 2019).

Human OT research faces methodological challenges. Some studies entail intranasal OT administration to test for potential causal effects of OT on human social behavior (Quintana et al., 2020), whereas others assess OT change in relation to a stimulus thought to elicit OT increases. Studies measuring acute human OT responses rely on different biological samples (blood, urine, saliva) to quantify peripheral hormone concentrations (Lefevre et al., 2017); typically control for sex differences (e.g., Kekecs et al., 2016); and involve a variety of study settings (i.e., lab versus naturalistic environment; e.g., Powell et al., 2019). As noted by Powell et al. (2019), these variations make it difficult for cross-study data comparisons, even as different approaches (e.g., intranasal oxytocin administration vs. salivary oxytocin measurement) offer respective advantages and disadvantages such as potential causal insight and ecological validity.

Building on this background, we sought to investigate changes in peripheral OT levels in reproductive aged women following an interaction with their pet cat. Another objective of this study was to quantify human and cat behaviors occurring during these sessions to evaluate possible associations between OT responses and the type of human-cat interaction. As such, we tested the following hypotheses:

H1: Women’s peripheral OT levels would increase after interaction with a pet cat, relative to a control condition.

H2: Changes in women’s OT levels during a cat interaction would be positively correlated with “maternal” behaviors directed towards their pet cat.

H3: Changes in women’s OT levels during a cat interaction would be positively correlated with cat affection-seeking behaviors, but negatively with cat anti-social behaviors.

Materials & Methods

Recruitment & participant screening

Study procedures were reviewed and approved by the University of Nevada, Las Vegas Biomedical Internal Review Board (Project 1490635-4). Participants were recruited in Las Vegas, Nevada, USA via word of mouth, flyers, and social media between December 2019 and December 2020. Recruitment materials described the study as an academic project to evaluate OT’s role in the bond between women, aged 18–45 years old, and their pet cat. The study also advertised participants would be compensated with a $20.00 Amazon gift card after they completed the study. Interested participants were directed to complete an initial screening survey using Qualtrics (http://www.qualtrics.com). The survey started with the informed consent document before participants could start the survey. Once participants provided informed consent, the survey began to screen participants and enroll eligible participants.

The screening survey consisted of questions to confirm participants met the following criteria: female, between 18–45 years of age, worked at least 30 h per week away from home, had been employed for at least six months, and owned their cat for at least six months. Due to the study’s purpose of evaluating OT, participants who were pregnant, lactating, menopausal, or had children under the age of 8 years old were excluded from participating. We recruited females of reproductive age as a result of previous research related to OT’s sex-specific effects (Petersson, Lundeberg & Uvnäs-Moberg, 1999; Carter, 2017) which we explain further in our discussion section. Participants meeting inclusion criteria were directed to complete additional questions about their sociodemographics (e.g., marital status, education level), their cat’s demographics (e.g., sex, age), and other pets in the household.

Participants

As shown in Table 1, we recruited 30 female participants between 19 and 41 years of age. The relationship status of our participants included single (n = 8, 26.7%); in a relationship and living apart (n = 6, 20%); in a relationship and cohabitating (n = 8, 26.7%); and married and living together (n = 8, 26.7%). Two women had children over the age of 8 and ten women reported being on hormonal birth control. The majority of participants only owned pet cats (n = 22), six had dogs, and three had other small, exotic pets (i.e., aquarium fish, unspecified turtle species, unspecified lizard species, unspecified snake species). Of the six participants with dogs and cats, two favored their dog over their cat. Seventy three percent identified as a cat parent (n = 22), 10% as cat friend (n = 3), 10% as cat guardian (n = 3), and 7% as cat owner (n = 2). Most participants obtained their cat from a shelter (n = 12, 40.0%), while others obtained their cat as a stray (n = 7, 23.3%), from a breeder (n = 1, 3.3%), from friends or family (n = 3, 10.0%) or from unidentified means (n = 7, 23.3%). All participants reported themselves as the primary caregivers for their cat and their cats as living indoors only.

Table 1 Descriptive participant sociodemographic characteristics.

Variable	N	Mean (SD) or %	
Age	30	29.7 (6.6) years	
Sexual Orientation			
Heterosexual	23	76.7%	
Lesbian	2	6.7%	
Bisexual	5	16.7%	
Ethnicity			
White	16	53.3%	
Hispanic or Latina/x	8	26.7%	
Asian or Asian American	2	6.7%	
Biracial or Multiracial	4	13.3%	
Relationship Status			
Single	8	26.7%	
In relationship, live apart	6	20.0%	
In relationship, live together	8	26.7%	
Married, live together	8	26.7%	
Household Demographics			
Live alone	6	20.0%	
Live with other roommates	3	10.0%	
Live with significant other	12	40.0%	
Live with family	9	30.0%	
Number of Household Pets (Cats)			
1	5 (7)	16.7% (23.3%)	
2	11 (15)	36.7% (50.0%)	
3	9 (5)	30.0% (16.7%)	
4	2 (2)	6.7% (6.7%)	
5+	3 (1)	10.0% (3.3%)	
Educational Attainment			
High school or equivalent	2	6.7%	
Some college	12	40.0%	
Bachelor’s degree	7	23.3%	
Graduate degree	9	30.0%	

Procedures

Following the screening survey, researchers contacted participants to review study procedures and schedule the dates for the participants to take part in the study. Two sampling dates were scheduled on the same week day and were spaced one week a part. One day of sampling consisted of the control (reading a book) and the other involved the participant interacting with their cat. Participants were not informed which date each interaction was to occur on, only that they would participate in both interactions with one occurring on one date and the remaining interaction on the follow-up visit. Participants were not made aware of the treatment until the day of their sampling. The sequence of the interactions was varied by pre-determined random but equal distribution of the control and treatment conditions to avoid order effects. Fifteen participants read the book on their first week and interacted with their cat on their second week while fifteen different participants interacted with their cat on their first week and read the book on their second week. Participants were instructed that on the days they were to participate they were to have been working or performing a work-related task away from their residence for a minimum of 4-hours and not in contact with any nonhuman animal during that time period. Participants were also instructed to not eat or drink one hour prior to their participation to prevent saliva sample contamination.

Additionally, this study took place directly after the government imposed restrictions due to COVID-19. To minimize in-person interaction before starting the study, additional steps were taken to eliminate researcher-participant interactions. During the scheduling process for each interaction, researchers emailed participants with day-of study protocols and an instructional video showing participants how to collect their own saliva sample. Researchers contacted the participant on the day of study to confirm participation, ask participants if they felt they had symptoms of COVID-19, and to inform the participant of the type of interaction (i.e., book reading or cat interaction). Researchers arrived at the participant’s residence before the participant to drop off the study materials (e.g., saliva collection tubes, sample storage cooler) in a location as designated by the participant. Participants were instructed to collect the first sample as soon as they arrived home, complete the 15 min interaction, collect the second sample, and then call the researcher to pick up materials.

Sampling occurred between September 2020 and January 2021. On the day of each participation event, participants supplied two saliva samples: one pre-interaction and one post-interaction (two samples for each day, four saliva samples in total). For each saliva sample, participants provided four mL of passive drool by filling up a five mL polypropylene tube to the designated mark and closing the tube with a tight-fitting polyethylene cap. As soon as the participant arrived to their residence, they collected the study materials, and supplied the first saliva sample. Once arriving home for each interaction, participants were not to interact with other people or non-cat pets. The participants were instructed to avoid contact (visual, auditory, or tactile) with their cat(s) and not to read book pages until after pre-test measurements were completed. Participants were also instructed to collect their post sample approximately 15 min after each interaction. Previous research suggests OT concentrations peak between 10–15 min after exposure (De Jong et al., 2015).

The control condition involved the participant reading a neutral, nonfiction book titled Roadside History of Nevada (Moreno, 2000), starting with page 119, for 15 min. Participants were instructed to read the book pages without distraction (visual, auditory, or tactile) and in a separate room from other individuals or pets. If a separate room was not available, participants were allowed to utilize their personal vehicle outside of their home. The cat interaction involved participants interacting with their cat(s) for 15 min. During the cat interactions, participants followed instructions to interact with their cat(s) as they would normally after returning home from work which included talking, petting, and/or playing with their cat(s). Participants were asked but not required to interact with one cat but due to living situations many were not able to segregate their cats. Cat interactions were to be recorded with a video recording device of the participants’ choosing (e.g., personal cell phone). No instructions were provided on how to record the interaction, only that the participant was to provide audio/video recording of their cat(s), and to attempt to focus the recording on one cat if there was more than one cat in the interaction. At the end of each 15 min session, the researcher notified the participant via phone to collect the second sample. Participants were advised to not read or interact with their cat when providing the second sample. At the end of each sampling session, samples were collected from each resident in a personal cooler and then transported to be stored in a −20 C chest freezer. Participants were instructed to upload their cat(s) interaction video to a platform of the participants choosing (e.g., google drive, dropbox) and share a link with the researcher at the end of each cat interaction. Once all samples and videos were collected, samples were shipped to the University of New Mexico for assays in February 2021.

Measures

After the second (final) sampling date, participants completed a follow-up survey. Participants were asked about the primary cat they interacted with and if they worked the day of each interaction. All participants indicated they followed work related instructions. Participants were asked if a child may have interacted with them during any of the interactions, though all participants answered “no” to this question. The intent of these questions were to assess whether the participant followed instructions, or if researchers would need to repeat the interaction at a different date. Additional questions included participant self-reported stress and interest during each interaction. Participants were asked to rate their stress on a Likert scale that ranged from 0 (“not at all”) to 10 (“extremely stressful”). Participants’ mean scores and standard deviation for stress on both interaction days were M = 5.47, SD = 2.30 (stress on the day of cat interaction), and M = 3.33, SD = 1.37 (stress on the day of book interaction). Participants also rated their interest level for each interaction (book and cat) on the same Likert scale. Participant mean and standard deviation interest scores were M = 9.00, SD = 1.15 (cat condition) and M = 3.77, SD = 2.47 (book condition). Though a limitation to our study, we did not ask participants to report stress or interest scores prior to and after each interaction and cannot correlate them to OT data. Participants were also asked to complete optional questions about relevant health aspects (e.g., medications) that might impact study results; however, many participants did not complete the optional questions and the questions were excluded from analysis.

Table 2 Ethogram describing the behaviors observed in the present study.

Coded Behavior	Sampling Method a	Definition	
Affectionate Interaction	
Human hugs/kisses cat	M	Participant hugs or kisses cat (does not imply cat consent)	
Gentle petting	S	Human makes gentle physical contact with their cat i.e., soft petting, cuddling	
Gentle skin to skin beyond petting	S	Human makes gentle skin to skin physical contact with skin that was not humans hands i.e., human rubs face on cat	
Gentle or baby voice	S	Human speaks to cat gently, including baby voice/whispering:	
Direct voice	S	Human speaks to cat in direct voice (as if speaking with another human in general conversation)	
Scolds or harsh voice	S	Human speaks to cat scolds/harshly	
Stimulatory Behavior	
Human attempts to play	M	Human throws toy or attempts to play with cat	
Human and cat play together	S	Human and cat play together	
Groomed with brush	S	Human grooms cat with a brush	
Affection-seeking Behavior	
Human-initiated contact	M	A human action that initiates physical contact (e.g., human reaches out their hand and pets the cat)	
Cat-initiated contact	M	A cat action that initiates physical contact (e.g., cat pushes head into human hand)	
Cat displayed affiliative behaviorb	S	“Friendly” behaviors that may communicate the cat’s intention to associate with other individuals in a peaceful manner.
Base Behaviors include: Follow, Gurgle, Head butt, Huddling, Lick, Nuzzle, Play, Prusten, Puff, Play Roll on Back, Groom/Allogroom, Rub/Allorub, Touch noses, Stutter	
Cat purredb	S	Low, continuous rhythmical tone produced during respiration while the cat’s mouth is closed. Creates a murmuring sound.	
Conversation with cat, paused for response time	B	Human speaks to cat and pauses for a response for over 5 s.	
Anti-social Behavior	
Cat displayed aggressive or agonistic behaviorb	S	Offensive behaviors communicate an intent to cause injury or engage in physical combat. Hostile behaviors associated with the confrontation with owner or other cats. (not play behavior) Base Behavior include: Attack, Arch Back, Avoid, Bare Teeth, Bite, Charge, Chase, Crouch, Cuff, Displace, Ears Back, Ears Flat, Fight, Flee, Ground slap, Growl, Pounce, Raise Paw, Rake, Retreat, Snap bite, Snarl, Spit, Stair, Strike at, Tail Slap, Tail twitch, Tail Under, Yowl	
Focused on another cat in environment	B	Human focuses on another cat in the environment that was not a focal cat	
Notes.

a Coding of observed human-cat behavioral interactions was separated into three categories: binomial, within a minute of each minute, and by seconds. Binomial (B) sampling required observers to provide a yes or no response if the behavior happened at all during the interaction. Binomial per minute (M) included watching the recorded video and hash marking a behavior/event one time with-in each minute of the video. For example, min. 1 the behavior happens, it is marked down, min. 2 the behavior happens again, it is then marked again, and if the event does not happen then it is not marked in that specific minute. Sampling by seconds (S) included watching each member of the interaction and writing down behaviors in seconds. Some of these behaviors have non-continuous streams thus any behavior with a 10 s pause was stopped and once it began again it was resumed. If a behavior started and resumed in less than 10 s, then the behavior was considered to be the same continuous activity observed.

b Ethograms based on a standardized ethogram for the Felidae: A tool for behavioral researchers (Stanton, Sullivan & Fazio, 2015) was utilized to define cat behavioral categories.

The video recordings from participant-cat interactions were reviewed to quantify both participant and cat behaviors. While no instructions were included to participants on how to record the cat interaction, most participants held their camera directly towards the cat, and some set their cameras against an object near the individual and cat. All focal cats were audible or visible during the entirety of the interactions. On average cats were visibly participating in the interaction for 13 min and 15 s. Due to the exploratory nature of this study and the naturalistic setting, this measure was not accounted for in behavioral coding or analysis. As shown in Table 2, participant and cat behaviors were defined and placed in specific categories to test our second and third hypotheses. Maternal-infant behaviors were adapted from Feldman and colleagues (2010), including affectionate touch (e.g., hugging and kissing, gentle petting, and skin-to-skin contact beyond petting), and from literature regarding human vocalizations (e.g., gentle or baby, direct, scold or harsh tones; Feldman et al., 2007; Seltzer, Ziegler & Pollak, 2010). Stimulatory behaviors were adapted from Feldman and colleagues (2010), including exploratory behavior (e.g., presenting objects, and re-directing attention to objects such as playing with a toy or grooming with a brush) to control for behaviors that are not linked to maternal OT increase in parent-infant literature. Cat behaviors such as vocalizations (e.g., purring), affiliative (e.g., grooming), and aggressive/agnostic behaviors (e.g., hissing) were gathered from the standardized felid ethogram developed by Stanton and colleagues (2015), and were part of our observed affection-seeking behaviors, and anti-social behaviors. Participant initiates contact, cat initiates contact, and participant had a conversation with their cat and paused for a response were conceptualized from both human and animal literature (Turner, 1991; Atzil, Hendler & Feldman, 2011; Barrett & Fleming, 2011; Uvnäs-Moberg, Handlin & Petersson, 2015). Much literature also comments on the dyadic nature of a relationship between a human and their infant or a human and their dog. Due to participants (n = 23) having more than one cat in the interaction, we included additional behavior variables (e.g., Participant focuses on another cat in the room) to provide insight on this aspect of a dyadic relationship between the human and focal cat during the interaction, or split attention by the participant as represented in the focuses on another cat variable which may affect bond formation during the interaction.

To maximize accuracy within behavioral coding, two observers trained on randomly selected recordings from the study sample and compared notes prior to finalized behavioral coding. Inter-observer reliability assessments were established by evaluating each behavioral variable, as shown in Table 2, within three video recordings. To evaluate inter-observer reliability, intra-class correlations (ICCs) were calculated (Rousson, Gasser & Seifert, 2002; Rousson, 2011) and were greater than 0.93. Once inter-observer reliability was established, the remaining video data were coded. ICCs for variables in all 30 videos remained above 0.93. To provide one unit for each behavioral variable measured in seconds, the difference between each set of behavioral variables was divided and added to the lower score. Scores were then rounded to whole numbers. Each behavior was measured in one of three ways (see Table 2): by either one of two frequency counts (e.g., binomial per minute or binomial overall) of the occurrence of a target behavior during the video, or by a continuous measure (e.g., seconds) of the duration of a behavior. For instances in which there were differences between observers’ coding of behaviors measured by frequency counts, both observers re-evaluated video content and the scoring criteria to come to an agreement. Descriptive characteristics of behavioral measures are shown in Table 3.

Table 3 Descriptive behavioral characteristics.

Coded Behavior	N	Mean (SD) or %	
Affectionate Interaction	
Human hugs/kisses cat	30	2.23 (2.81) BPMs	
Gentle petting	30	346.73 (307.94) seconds	
Gentle skin to skin beyond petting	30	36.03 (73.18) seconds	
Gentle or baby voice	30	196.23 (140.65) seconds	
Direct voice	30	60.57 (77.95) seconds	
Scolds or harsh voice	30	0.90 (2.06) seconds	
Stimulatory Behavior	
Human attempts to play	30	3.93 (4.14) BPMs	
Human and cat play together	30	131.73 (149.71) seconds	
Groomed with brush	30	29.53 (96.16) seconds	
Affection-seeking Behavior	
Human-initiated contact	30	5.67 (3.26) BPMs	
Cat-initiated contact	30	4.10 (3.25) BPMs	
Cat displayed affiliative behavior	30	549.83 (224.58) seconds	
Cat purred	30	230.80 (293.96) seconds	
Conversation with cat, paused for response time	11/19	36.7% /63.3%	
Anti-social Behavior	
Cat displayed aggressive or agonistic behavior	30	14.40 (31.13) seconds	
Focused on another cat in environment	5/25	16.7%/83.3%	
Notes.

BPM, (Binomial per minute).

OT levels were measured using commercial OT-ELISA kit (Enzo Life Sciences, Farmingdale, NY, Catalog #: ADI-901-153A) in the Comparative Human and Primate Physiology Laboratory at the University of New Mexico (directed by Melissa Emery Thompson) using methods previously reported (Grebe et al., 2017). Briefly, samples were thawed, vortexed thoroughly, and centrifuged for 15 min to break up and precipitate mucins. To reduce possible matrix effects and obtain sample concentrations within the range of kit standards, 1.5 ml of clean saliva was freeze-dried and reconstituted into 0.25 ml of sample buffer (6x concentration). Carter et al. (2007) reported robust recoveries of oxytocin from lyophilized samples, though this validation was performed with an older version of this assay kit. Using the current kit, Daughters et al. (2015) demonstrated parallelism of reconstituted saliva with assay standards. While sample concentrations of 2.5x-6x are reported in prior studies, we note that full analytical validations have not been conducted for the 6x concentration against this antibody. This assay, which has been validated by the manufacturer for use with human saliva, has a sensitivity of 15 pg/mL (90 pg/mL for our concentrated samples). The mean intra-assay coefficient of variation (CV) was 9.8% for duplicate determinations of our samples. The inter-assay CV was 14.8% for a low-concentration control and 5% for a high-concentration control. The manufacturer reports negligible cross-reactivities (<0.02%) to related mammalian peptides, such as vasopressin.

Statistical analyses

The Statistical Package for the Social Sciences (SPSS, version 25.0, IBM software) was used for statistical analysis. For OT analyses, the Shapiro–Wilk test was used to determine the normality of the distribution. All OT results presented here rely on Log10 transformations due to non-normal distribution of raw data. The transformation of data was applied to make the data conform to normality, and increase the validity of further associated statistical analyses. This transformation has also been utilized in other OT studies specifically related to human-animal interaction (e.g., MacLean et al., 2017; Marshall-Pescini et al., 2019). OT values of log pre- and log post-conditions were used in a repeated measures ANOVA as within-subject variables to compare both the experimental condition and the control condition as between-subject factors and to consider whether potential covariates might yield different OT levels between participants.

The difference in OT concentration (established by subtracting the log-transformed post conditions value by the log-transformed pre conditions value) represents a ratio of the two log-transformed concentrations and was used in Spearman’s rank correlation coefficient (rho) to assess possible correlations between OT and behavioral observations. This non-parametric analysis was used because normal distribution was not assumed for the behavioral data. We restricted the behavioral analyses to the cat condition because of the lack of women’s difference in OT change by condition and because of the study focus on evaluation of potential effects of interacting with a cat. An alpha value of 0.05 was considered statistically significant in all analyses, unless otherwise noted.

Results

Literature suggests sociodemographic characteristics are potential covariates that might yield differences in OT levels between participants. For our study, hormonal contraceptive was not considered as a covariate due to the limited number of participants using birth control and the lack of detailed information about hormonal birth control formulations. Additionally, participants’ relationship with their cat was not considered as a covariate due to the high number of participants identifying as a cat parent. Only two women had children over the age of 8 years old and, as such, having children was not considered a covariate. Univariable analyses were used to test whether log OT differed by participants’ education and relationship status (e.g., single or partnered but living apart). For log pre- and post-cat conditions, education and relationship status were not significant (F (3, 25) = 0.046, p = 0.986 & F (3, 25) = 0.082, p = 0.969, respectively) and neither were the log pre- and post-control conditions (F (3, 25) = 2.135, p = 0.121 & F (3, 25) = 0.674, p = 0.576, respectively). Accordingly, we subsequently employed univariable analyses to test human OT change in response to each interaction.

Mean values of women’s OT levels for the cat condition and control condition included untransformed data (pre-cat condition M = 18.73, SD = 12.81 pg/ml, post-cat condition M = 19.44, SD = 11.06 pg/ml, pre-control condition M = 19.81, SD = 10.02 pg/ml, post-control condition M = 18.16, SD = 8.73 pg/ml) and log-transformed data (pre-cat condition M = 1.17, SD 0.31 pg/ml, post-cat condition M = 1.22 SD = 0.25 pg/ml, pre-control condition M = 1.24, SD = 0.23 pg/ml, post-control condition M = 1.20, SD = 0.26 pg/ml) (see Table 4). Across conditions and timepoints, means were relatively similar for both untransformed data and log-transformed data and due to non-normal distribution of the adjusted concentration data we ran repeated measures ANOVA on log-transformed data (see Fig. 1).

Table 4 Mean, standard deviations, and range of OT levels in pg/mL.

Mean ± SD (range)	
	Pre-Cat	Post-Cat	Pre-Control	Post-Control	
OT levels	18.73 ± 12.81
(2.72–55.48)	19.44 ± 11.06
(4.82–48.63)	19.81 ± 10.02
(4.62–45.45)	18.16 ± 8.73
(2.78-36.18)	
LOG OT levels	1.17 ± 0.31
(0.43–1.74)	1.22 ± 0.25
(0.44–1.56)	1.24 ± 0.23
(0.66–1.66)	1.20 ± 0.26
(0.44-1.56)	

Figure 1 LOG Oxytocin levels in women before and after interaction with their cat are on the left and LOG Oxytocin levels in women before and after the reading condition are on the right.

Each line represents one person. Increases are shown with a ▴ and decreases are shown with a ■. Between the pre-cat and post-cat 17 participants OT increased and 13 participants OT decreased. Between the pre-book and post-book interaction 15 participants OT increased and 15 participants OT decreased.

The repeated measures ANOVA was used to answer whether the mean change in the outcome from pre to post differed in the two groups. No statistically significant difference was found (F (1, 58) = 0.003, p = 0.954). Additionally, we found no statistically significant between pre-post OT by condition interaction effects (F (1, 58) = 2.308, p = 0.134). Accordingly, these findings do not support our hypothesis that women’s interactions with a pet cat would increase women’s OT levels, relative to a control condition (book reading).

Thereafter we tested the difference in women’s OT concentration from our cat condition in a Spearman’s rank correlation coefficient to assess possible correlations between OT and behavioral observations. As shown in Table 5, many of our results support our second hypothesis that participant maternal behaviors (kissing and hugging, petting, and skin-to-skin contact beyond petting) were positively correlated with increases in women’s OT; however, there were no associations between women’s OT and vocalization measures. These findings mean that women’s OT changes during the interaction reflect specific mother-infant interactions.

Table 5 Spearman’s rho significance and correlation coefficient of the difference in women’s OT concentration with behavioral data for (N = 30) participants during the cat condition.

Spearman’s rho	Items correlated with women’s OT difference during the cat condition	Correlation coefficients (r)	Sig. (p)	
	Affectionate Behavior	
	Human hugs/kisses cat	0.419	0.021	
	Gentle petting	0.471	0.009	
	Gentle skin to skins contact beyond petting	0.377	0.040	
	Gentle or baby voice	−0.080	0.676	
	Direct voice	−0.290	0.120	
	Scolds or harsh voice	−0.206	0.275	
	Stimulatory Behavior	
	Human attempted to play	−0.206	0.275	
	Human and cat play together	−0.202	0.285	
	Groomed with brush	0.172	0.362	
	Affection-seeking Behavior	
	Human-initiated contact	−0.096	0.615	
	Cat-initiated contact	0.562	0.001	
	Cat displayed affiliative behavior	0.487	0.006	
	Cat purred	0.131	0.282	
	Conversation with cat, paused for response time	0.555a	0.001	
	Anti-social Behavior	
	Cat displayed aggressive or agonistic behavior	−0.453	0.012	
	Focuses on another cat in environment	−0.377	0.040	
Notes.

a Reverse coded.

To test our third hypothesis, we ran the non-parametric correlations between the difference in women’s OT concentration during the cat condition with variables obtained from behavioral observations. For our third hypothesis, the difference in women’s OT concentration was positively correlated with most of our affection-seeking behavioral measures (cat-initiated physical contact, cat displayed affiliative behavior, and participant had a conversation with cat, and provides a response time), though not all (unrelated to human-initiated contact and cat purred). Our anti-social measures were both significantly negatively correlated with the difference in women’s OT concentration. These findings mean that the differences in women’s OT during the interaction are associated with both behaviors of the participant and their cat.

Discussion

This study makes a novel contribution to research investigating OT’s role in human-animal interactions and how the types of behaviors occurring during these interactions influence peripheral OT levels. Our results showed no differences in OT change between the control condition and cat interaction. Results also provided evidence that women’s OT responses were correlated with specific behaviors during the cat interaction such as gentle petting, hugging and kissing and cat-initiated contact but not correlated with behaviors such as gentle or baby voice, cat purred, or human-initiated contact. Both of these findings are inconsistent with Curry and colleagues’ (2015) findings in which women’s OT levels decreased following unfamiliar cat interactions. In moving forward, this research not only shows the importance of accounting for the use of unfamiliar and/or familiar animals in human-animal studies but also of evaluating specific behaviors rather than or in addition to an unfamiliar/familiar cat interaction condition. Further research should place more value on the variable behaviors within the human-animal interaction rather than exclusively the animal’s presence.

Due to our predetermined sample size, previous research related to OT’s sex-specific effects (Carter, 2017), and major endocrine changes in puberty, pregnancy and menopause, we specifically sought to recruit females of reproductive age (between 18–45 years of age) and exclude males in an attempt to reduce the potential for varied findings. More specifically, in animal models previous research establishes that estrogen, a stimulus for OT production and secretion, maintains longer lasting effects in females as compared to males (Petersson, Lundeberg & Uvnäs-Moberg, 1999). One of our original intentions in this study was to understand the impact interacting with a pet cat may have on stress reduction after a full day at work as the workplace is a potential source of stress. We designed the study to conduct interactions that better simulate a real-life experience in which the participants would arrive home and interact with their cat after a day of work (see Miller et al., 2009). However, our study was modified and subsequently followed the government-imposed social isolation with an ease in restrictions in which many individuals worked from home. While we continued to exclude participants that worked less than 30 h per week, and who had not been employed for at least six months for consistency, we only required each participant to be working away from their cat and home for 4 h prior to the interactions to separate the potentially stressful work environment and the potential stress reducing effects of interacting with their cat to make an interpretation of OT results more uniform between participants. Additionally, we excluded women who had children under the age of 8 years old as they are typically more dependent on their caregiver, and need continuous supervision which would not be ideal for this study. Accordingly, the participants in this study can be characterized as highly-invested, working, reproductive-aged US “cat moms,” few of whom had human children.

A recent study indicates pets serve as an outlet for nurturing, distraction and social support while observing government-imposed social isolation during the COVID-19 pandemic for individuals living in homes without children (see Johnson & Volsche, 2021). This research may speak to the quality of interpersonal relationships that women within our sample may have had with their cat. One of the criteria for the study was that individuals have their pet cat for more than 6 months at the time of initial survey completion, which aligns with the timeline of Johnson & Volsche (2021). OT levels may be affected by the quality of interpersonal relationships which also may explain the differences in findings between our study and Curry and colleagues (2015). Thus, we believe our sample was a human sample for whom one might be most likely to observe an increase in women’s OT during interactions with their pet cats, particularly when compared in cross-cultural contexts for which cats are more commonly valued for their roles in vermin removal than as furry family members (Gray & Young, 2011).

Previous studies support the notion that human-dog and human-cat interactions are similar to maternal-infant interactions (Stoeckel et al., 2014; Edwards et al., 2007, respectively), while only one study has considered human and cat (unfamiliar cat) interactions as a stimulus for human OT response (Curry et al., 2015). OT has been well-documented in maternal bonding and behavior, including skin-to-skin contact, positive affect, and affectionate language (Kendrick, 2000; Bakermans-Kranenburg & van IJzendoorn, 2008; Gordon et al., 2010; Bell, Erickson & Carter, 2014). Our research supports the idea that tactile maternal type affection also occurs in human-cat interactions and is positively associated with the difference in women’s OT concentrations. However, we found no correlation between the difference in women’s OT concentrations and various vocalizations. Another relevant study found that touching an animal produced greater outcomes in stress reduction than speaking to an animal (Vormbrock & Grossberg, 1988). Touch is an essential part of communicating information in early development for many animals (Cascio, 2011) and may also play a role in OT production. Due to social conventions, individuals often engage in more touch when interacting with animals as compared to other human individuals. Additionally, many colleges and universities in the United States have programs involving human-animal interactions as a means to reduce stress and improve mental health in which the participant is not familiar with the animals (Crossman, Kazdin & Knudson, 2015). It is possible to speculate that individuals are potentially gaining OT-related benefits from interactions with unfamiliar animals and even objects (review Peled-Avron, Perry & Shamay-Tsoory, 2016; Geva, Uzefovsky & Levy-Tzedek, 2020). An alternative interpretation of our findings draws upon previous research suggesting tactile stimulation may create an animal-specific benefit of grounding comfort or distraction (Bleiberg et al., 2005; Beetz et al., 2012) that contributes to increased OT and stress-reducing benefits. On the contrary, it is possible that both bonding and tactile stimulation play a role as suggested in human-dog interactions mentioned earlier and in the comparison of our study to Curry and colleagues (2015). Further research should consider including control conditions that provide tactile stimulation (i.e., furry objects) to assess if OT levels respond to pet cats or general tactile stimulation.

This study also examined play behaviors and grooming with a brush and found no correlation with OT. This may be explained by parental OT literature, in which maternal OT is associated with affectionate care while paternal OT is correlated with stimulatory, proprioceptive, and object-oriented play (Gordon et al., 2010). Both men and women may derive higher rewards from specific behaviors, but we could not test this due to our study design. Future studies should include men to compare OT response to pet cat interactions.

Looking at maternal-infant behavior, an fMRI study suggests synchronicity of behaviors (e.g., coordination between maternal behavior and infant signal) correlated with maternal OT levels, while mothers who displayed intrusiveness (e.g., excessive expression of maternal behavior) in the interactions lacked correlations with OT levels (Atzil, Hendler & Feldman, 2011). Turner (1991) conducted research to measure human-cat relationship success and found a negative correlation between women initiating interactions and the length of interaction time of the human-cat interaction, suggesting human-cat interactions last longer when initiated by the cat. In comparison, Turner’s study (1991) was unrelated to OT but addressed coordination and compliance between a human and their cat. Our study observed both cat-initiated contact and human-initiated contact and conversely found no correlation between human-initiated contact, but we found a correlation between increased positive OT and cat-initiated contact which may suggest hormonal synchrony between the dyad. This observation aligns with more recent evidence suggesting that OT may be related to the perception of a situation rather than an action of stroking and may also be animal specific (Nagasawa et al., 2015; Lürzel et al., 2020). Nagasawa and colleagues (2015) found that OT concentrations were affected by dog non-verbal communication (gaze and touch), and reported evidence of a self-perpetuating oxytocin-mediated positive loop similar to mother-infant bonds. MacLean and colleagues (2017) found that increases in dog OT were predicted by the extent of affiliative behavior in a human-dog interaction.

We found an additional significant correlation between cat behaviors and women’s OT, including cat displayed affectionate behavior and cat displayed agnostic or aggressive behavior. Behavioral synchrony in human-dog literature is often associated with an increased affiliation or social responsiveness in dogs with their adult owners (Wanser, MacDonald & Udell, 2021). Therefore, our affection-seeking and anti-social measures may contribute to behavioral synchrony or asynchrony similar to human-infant and human-dog synchrony, which account for neurobiological systems in social bonding and attachment that impact women and their relationship with their pet cats. Additionally, anti-social behaviors observed in human participants, such as the focus on another cat, may break the relationship’s dyadic nature, which may explain the negative correlations with women’s OT.

Regarding communication measures and the difference in women’s OT concentration, we found a correlation on a specific discussion type in which women spoke to their cats and paused for the cats’ response. Often, participants spoke to their cat in the same manner someone would speak with a baby or young child, but if a participant spoke to their cat and waited for their cat’s response (e.g., meow, touch, attention) their action was correlated with a positive increase in OT. Pet-directed speech (PDS), represented in this study, is considered to be a type of speech characterized by a high-pitched voice, a wide pitch range, slow rate, simple syntax and semantics, and high repetition of words compared to more neutral adult-directed speech (ADS) (Singh, Morgan & Best, 2002; Ben-Aderet et al., 2017). PDS shares similar structural properties to infant-directed speech (IDS) which captures and maintains infant attention and facilitates social interaction (Cooper et al., 1997; review Saint-Georges et al., 2013). Recent studies found that dogs and horses are also sensitive to PDS which increases their attention and gaze towards humans compared to ADS and promotes interspecies communications (Ben-Aderet et al., 2017; Lansade et al., 2021). Considering cats also have social cognitive abilities (Vitale, Behnke & Udell, 2019) and the ability to use human cues (Chijiiwa et al., 2021), they likely respond similarly to PDS. Therefore, we suggest that cat reactivity or interactive feedback to human speech in this study demonstrates an interactive loop to caregiver solicitation, which may provide women with emotional responses like that of mothers to their infants, and increases OT production to promote social bonding formation between species. Because IDS is considered a caregiving behavior, it can be associated with an increase in OT levels in both mothers and their infants when mothers are providing care for their infants (Chisholm et al., 2005). It is possible that when PDS is used and pets respond to their caregivers, both parties’ OT levels increase. While this was not the aim of our study, our research does open up new research avenues for future work for the impacts of PDS and OT increases in both cats and their care providers. Further research could also consider various facets of interspecies communication such as acoustic characteristics of voice, facial expression, context of discussion and emotional information presented during human-animal interactions and their effects on OT.

Limitations

One notable limitation of this study is the method used to examine changes in OT. Saliva provides a measure of peripheral OT levels, and the differences in central and peripheral OT releases are still unclear (Powell et al., 2019). This limitation applies to all human OT research as central OT is invasive requiring cerebrospinal fluid, while peripheral can be measured in blood, urine, or saliva (Lefevre et al., 2017). Saliva was used for this study as it did not require professional assistance (i.e., phlebotomist) and causes minimal stress for participants. Saliva collection was also more conducive to participant sampling in a short amount of time. Some steps within the oxytocin assay protocol have not been validated, though this is advised (MacLean et al., 2019). Despite the limitations of peripheral OT as a measure, our findings are discordant with documentation of statistically significant increases in OT concentration in previous human-dog and human-infant research (e.g., Odendaal, 2000; Handlin et al., 2012; Miller et al., 2009; Nagasawa et al., 2009; Nagasawa et al., 2015; Powell et al., 2019), but concordant with a more recent study on salivary OT in human-dog interactions (see Powell et al., 2020).

At present, there are no standardized methodologies for evaluating human-animal interactions on physiological processes. Participants interacted with cats for 15 min, and most other studies have varying times between 5 and 30 min. Our study also lacked strict directions on video recording which can be seen as a limitation. This study also had many covariates and a small sample size. Few empirical studies with large sample sizes exist concerning human-dog interactions and OT. To further support this study, a larger sample size would be needed. In addition, several human-specific (e.g., medication, depression, anxiety) and cat-specific (e.g., neuter status, age) were not controlled for and may have impacted the findings.

Another limitation is related to our inability to directly observe participant interactions. To best account for this, participants were extensively screened and were provided detailed instructions on what to do and not do during their participation. Arguably, video recordings during the cat interactions provide insight into participant compliance. However, the participant was in control of what was recorded and does not account for what is occurring “off camera.” We also addressed this limitation by reminding participants of instructions the day of each interaction and by administering a follow-up survey about their participation. Future research can address this limitation by performing their study in clinical settings, though this research will also be limited by confounding factors such as the impacts of unfamiliar circumstances to participant stress levels.

Conclusions

Our research contributes to the growing data on human-animal interactions, to a large body of interdisciplinary research on OT and social behavior, and more specifically on patterns in women’s OT responses during interactions with their pet cats. While there was no univariable support for the hypothesis that women’s OT would increase during interaction with pet cats relative to the control condition, there was evidence that women’s OT responses were correlated with specific behaviors (such as human affectionate behaviors directed toward their cats) during the interaction. Pet ownership is a multifaceted relationship and consists of various components between both the owner and the animal that influence OT and human health; however, evidence on these mechanisms is sparse (see Herzog, 2011, for review). Much research still needs to be done to understand the role that OT plays in interspecies bonding. Future research should explore the role of relationship quality and interaction types on OT release, including human-cat social referencing, hormonal synchronization, and factors driving inconsistent findings. It is also essential to research and create tools that provide information to cat owners about various cat communication styles and behaviors to promote a healthy human-cat bond.

Supplemental Information

Supplemental Information 1 Demographic information, oxytocin results and behavior observation dataset

Click here for additional data file.

This paper and the research behind it would not have been possible without the assistance of Dr. Melissa Emery-Thompson, who oversaw the sample measurement in the Comparative Human and Primate Physiology Laboratory at the University of New Mexico and provided feedback on our manuscript. We want to thank Dr. Shelly Volsche for her encouragement to pursue this project and her feedback on this project design and statistical analysis. We also thank the UNLV Evolution and Human Behavior reading/lab group members for input on the research design and manuscript revisions, as well as the participants and their cats, who generously provided their time for this project.

Additional Information and Declarations

Competing Interests

Author Contributions

Human Ethics

Data Availability

The authors declare there are no competing interests.

Elizabeth A. Johnson conceived and designed the experiments, performed the experiments, analyzed the data, prepared figures and/or tables, authored or reviewed drafts of the paper, and approved the final draft.

Arianna Portillo performed the experiments, analyzed the data, authored or reviewed drafts of the paper, prepared the figures and/or tables and approved the final draft.

Nikki E. Bennett analyzed the data, authored or reviewed drafts of the paper, and approved the final draft.

Peter B. Gray conceived and designed the experiments, analyzed the data, prepared figures and/or tables, authored or reviewed drafts of the paper, and approved the final draft.

The following information was supplied relating to ethical approvals (i.e., approving body and any reference numbers):

Study procedures were reviewed and approved by the University of Nevada, Las Vegas Biomedical Internal Review Board (Project 1490635-4).

The following information was supplied regarding data availability:

The raw data and measurements are available as a Supplementary File.

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
