# Peer review of "Exploring women’s oxytocin responses to interactions with their pet cats"

_PeerJ, doi:10.7717/peerj.12393_

## Round 0.1 · original submission · Major Revisions

I was enthused to see you address human-cat interactions, which are understudied compared to human-dog relationships. I have been very fortunate to receive three helpful reviews. While two reviewers suggest only minor revision, based on their suggestions and those of the more critical Reviewer 3, I would like to invite a major revision. I am not an expert on hormonal analyses, so I defer to Reviewer 3's suggestion about the measurement of OT. Based on all of the reviewer comments together, it is clear that there are several methodological and design details that need clarification before the paper can be accepted.

You might want to cite Merola's work on social referencing in cats as well around line 87. Do not feel obligated to cite this but I believe there is an opportunity to integrate more of the existing work on human/cat interactions. There several other important papers on cat/human interactions that are not cited here such as work by Saito and Chijiiwa.

Why did participants have to be employed and work away from home?

Would it not have been better to have participants report stress before they took the first sample and again after the interaction to correlate with the OT readings?

I appreciate the apparently rigorous reliability coding but it is still a bit unclear as described. You should describe which behaviors were coded before describing reliability.

Can you indicate if the cats' behavior differed when they initiated the contact versus when the human initiated? I wonder if the difference here is not about who initiated but about the cat indicating more pleasure with the interaction when it was self-initiated?
How were the videos provided to the researchers?

Reviewer 1 ·

Basic reporting

Besides from some minor typo’s the paper is easy to read and not overly wordy. Relevant literature regarding the role of OT in evolution of mammal bonding, in human-infant bonding, human-dog bonding and different ways to measure OT is described.

In addition, categories of human-cat interactions and cat behavior are connected with knowledge from human-infant and human-dog interactions.

It is not clear to me if participants were instructed how to behave with their cat. If no instructions were given, the authors might add this to the method section.

I would like to see a table/graph to provide insight in the presence of the behaviors (% yes/no) and means (averages) of the other behaviours.

Experimental design

As OT is connected to social bonding, I was wondering why the following aspects were not considered as covariates: number of cats, self-reported bond with the cat and havein children (or not).

Validity of the findings

The three hypotheses are clearly answered and are (to the best of my knowledge) discussed in relation to what is known in this field. Limitations of the study are present.

Additional comments

Clear aim, hypotheses and design. Clearly written and relevant paper.

·

Basic reporting

Introduction

Line 79: I think "Though less clear" should be deleted. Does this expression imply that the results in Bell, Erickson, & Carter (2014) were not robust?

Paragraphs beginning with Line 83: You might consider citing Topál et al. (1998, Attachment Behavior in Dogs (Canis familiaris): A New Application of Ainsworth's (1969) Strange Situation Test). They demonstrated that dogs show patterns of attachment behavior toward the owner similar to mother-infant interaction in SST.

Lines 101-103: Consider citing Edwards et al. (2007, Experimental evaluation of attachment behaviors in owned cats) and Nagasawa et al. (2020, Effects of the characteristic temperament of cats on the emotions and hemodynamic responses of humans)

Line 118: I think the reason for limiting the experimental subjects to reproductive aged women is a little weak. Certainly, you have cited many papers about mother-infant bonding, but you also mentioned that “underlying neurological processes exist across both sexes and impact both male and female social cognition” in lines 70-71.

Lines:122-128: I think the three hypotheses are a bit abrupt. It would be better to explain in more detail whether maternal and affection-seeking behaviors are also observed in human and pet (dog) studies, although you mentioned that OT is implicated as promoting human-dog social interactions in line 88.

Experimental design

Materials & Methods

The procedure need a bit more explanation.
- In the control condition, did the participants read the book in a separate room so that cats or other pets could not come in?
- For the participants who have more than one cat, did you instruct them which cats they would interact with (e.g., the one kept the longest) ?
- When the participants record their interaction with cats, did they hold the camera (cell phone) by hand? Or did they put it on a tripod or something?

Measures
Line: 223: Participants were asked if they worked the day of interaction. It seems to contradict with the following statement: "Participants were instructed that on the days they were to participate they were to have been working or performing a work-related task away from their residence for a minimum of 4-hours" (lines 184-186).

Lines 231-232: Their stress score was higher on the day of cat interaction than the day of book interaction, is it right?

Lines 247-249: It would be better to have information about minute sampling in the manuscript as well as in the caption of Table 2.

Lines 268-269: If the presence of other cats could influence the participants' behavior, why didn’t you control it? Was it impossible to conduct in a room where no other cats could enter?

Statistical Analyses
Line 291: I apologize, but I am unsure why you conducted both ANOVA and ANCOVA. Could you please clarify?

Lines 322-324: A table or figure would be needed.

Line 328: If you test whether the mean change from pre to post differed in the two groups, why a t-test was not used? Or, did you conduct 2x2 design ANOVA? Did you mean there were neither main effect of condition (cat / book), order (pre / post) nor interaction of them?

Validity of the findings

Results

Lines 347-349: In Table 3, " Conversation with cat, paused for response time " seems to be negatively correlated with OT change, is that correct?

Discussion
Lines 369-370: Do you mean that even if not interacting with the animal, specific behaviors in themselves (for example, petting something) might correlate with OT responses?

Lines 378-381: Do you mean that an increase in women’s OT would not be expected by interacting with cats in another culture where cats are simply regarded as vermin removers?

Lines 384-385: There are a few studies about human-cat interactions that did not deal with OT (for example, Edwards et al. (2007) and Nagasawa et al.(2020) mentioned above), so it is better to clarify the differences between those studies..

Lines 408-410: consider citing Turner (1991, The ethology of the human-cat relationship), which suggested that cat-human interactions last longer when initiated by the cat.

Paragraphs beginning with Line 421: How about mentioning pet directed speech (PDS)?

Lines 438-439: “comparable studies with dogs and children have found effects that were not found in this study.” does this indicate that there was no difference in the change in OT between the cat and control conditions in the present study?

Additional comments

While the COVID-19 makes it difficult to conduct face-to-face experiments, it is nice that the participants were instructed how to conduct the experiment so that they could do it alone. As mentioned in the limitation section, this method does not allow you to directly observe the participants interactions, but it is important to continue research and explore new research methods even in such a difficult time.
It is interesting that women's OT changes were correlated with several behaviors during the human-cat interactions (e.g., positively with petting the cat and cat approach initiation, negatively with cat aggression behavior). However, it does not seem to be much discussion of the reasons why there was no difference in OT change between the control and cat conditions.

Reviewer 3 ·

Basic reporting

The methods are reported in good detail. The results are mostly presented clearly, but with an odd framing on the methods. The introduction and discussion are not clearly written, should cite more extensively, and the logical flow is often missing or implicit. I give specific recommendations in the general comments. I also suggest reading sections aloud to help find awkward sentences and shifts in topic.

Experimental design

The methods are well reported; however, I have two methodological concerns.

First, there is no analytical validation (parallelism and spike recovery) reported for samples lyophilized with a 6x concentration. Although another paper is cited for the method, I do not see a validation in that paper either. The kits themselves are validated on unaltered (unconcentrated, not lyophilized samples), however a concentration of 6x could easily concentrate other molecules that might interfere with the immunoassay. Thus, a parallelism should be conducted from above 6x (e.g. 8x) to ensure that the samples dilute in parallel with the standard curve. In addition, spike recovery should be assessed at the 6x concentration. (For more on these methods, see Andreasson et al. 2015.

Second, by taking the differences of already logged concentrations, you are essentially analysing the change ratio, rather than the absolute change in hormone levels. This is important to be clear about in presenting the results and this choice should be justified.

Andreasson, Ulf, Armand Perret-Liaudet, Linda J.C. van Waalwijk van Doorn, Kaj Blennow, Davide Chiasserini, Sebastiaan Engelborghs, Tormod Fladby, et al. “A Practical Guide to Immunoassay Method Validation.” Frontiers in Neurology 6, no. Aug (2015): 1–8. https://doi.org/10.3389/fneur.2015.00179.

Validity of the findings

The underlying data is provided, and the conclusions drawn generally seem valid. However, I am concerned regarding the way the correlations with OT change are presented (i.e. not as ratios), and also the mention of COVID-19 in the discussion that is not grounded in any reported data collection.

Additional comments

Introduction:
54-55: the use of the word “changes” makes sense to me as compared to the pre-cat interaction levels, but not relative to the control condition.
60: I am confused by “has establish there are components to behavior on”. Maybe “has identified distinct human and cat behaviors that influence oxytocin release in humans”?
68: I wouldn’t really consider birth (and particularly the physical aspect of contractions facilitated by OT) to be a social behavior. It seems more like a reproductive behavior.
83-84: I agree that this is a more common behavior in humans than other animals, but I’m uncomfortable with the use of unique here. There are certainly plenty of cases, both in the wild and in captivity, of strong bonds between members of different species.
85-86: “Domestic dogs possess social cognitive functions analogous to children”, this is either overclaiming or is open to misinterpretation as such. While dog social cognition does bear some resemblance to human children’s cognition (and it would be worth citing some examples), there are also clearly lots of limitations, and the analogies seem to apply in some respects (or on some tasks) more than others. Please clarify and be more specific.
88-90: What behaviors? Please expand on this. And what do you mean by “reflected biologically” other than oxytocin levels? Perhaps this could be simplified by just saying “Oxytocin has been implicated in X, Y, Z.”
92-99: this paragraph needs more structure. Guide the reader through your logic.
107: presumably you thought that an unfamiliar cat was meaningfully different from interacting with one’s own cat. Can you be more explicit about that here?
109-115: can you discuss the strengths and weaknesses of each approach? Just a few sentences.
The introduction generally focuses on the context of maternal behavior. Obviously this is the most relevant to the framing of the study, but it would also be worth mentioning the role of oxytocin in other social relationships, especially pair-bonding.

Methods:
148: why was the age of 8 years old chosen as the threshold?
181: missing a word (order? sequence?)
208: interaction, not intervention?
213: were videos set up to capture the interaction (e.g. on a tripod) or held continuously with focus on the cat? How much were the cats in frame? Assuming not the entire time, how was this treated in the behavioral coding and analysis?
267-269: I’m not entirely sure what you’re saying here. Are you trying to get at competition/jealousy between cats? Or split attention of the participant?
273: please provide the catalog number. This is especially important for immunoassays that are dependent on an antibody, which sometimes changes. Please also report the cross-reactivities of this kit with other analytes.
Table 2: are human-initiated and cat-initiated mutually exclusive? What rules do you use to determine? (E.g. what if the human reaches out their hand and the cat pushes their head into it?) “within a minute of each minute” I don’t understand (becomes clear later in caption, but this is confusing). Maybe binomial per minute?
288-289: It is more accurate to say “to check normality”, as a failure of Shapiro-Wilk could be for skew or for other reasons (and the test statistic does not tell you about skew).
295-296: subtracting logs is the same as taking the log of division, not subtraction. Thus, you area analysing a change ratio. This is important for the interpretation.

Results:
While it can be helpful to remind readers what you did (and why) as the beginning of the results, statements about the literature should be in the introduction or discussion, not the results. Furthermore, most of the results are introduced based on the test that produced them. I suggest reframing these sections to focus on the data/result/question, rather than the method. The results would benefit from a figure showing individual points and lines between pre and post. Is the change not large enough to rise to significance, is it there in most but not all individuals, or is it a pretty even mix or rising and falling? A figure that shows this would be helpful for the reader to interpret the negative result.
319: restate this to be about the subjects, not about the data.
324: I think “all” refers to across conditions and timepoints, but this is not entirely clear.
342-343: I am confused by this last sentence. Are you extrapolating these results to humans? Why?
348-349: two ands
350: to, not toe?
351: “suggest” or “indicate” not “mean”
Table 3: No indicators of correlation significance are actually given in the table. Instead of flagging certain variables as Y/N sampling indicating a positive correlation, I would suggest the reference level be swapped, which will lead to a more intuitive interpretation of the results.
364-366: explain what you mean by “inconsistent”. Unpack this. While the findings are different, these are pet cats, so that’s not surprising. What might it mean?

368-370: I agree that looking at specific behaviors is very important for behavioral endocrinology research! However, differential associations like this have been seen in other species as well, so please discuss and cite. For example:
Lürzel, Stephanie, Laura Bückendorf, Susanne Waiblinger, and Jean Loup Rault. “Salivary Oxytocin in Pigs, Cattle, and Goats during Positive Human-Animal Interactions.” Psychoneuroendocrinology 115, no. February (2020): 104636. https://doi.org/10.1016/j.psyneuen.2020.104636.
MacLean, Evan L., Laurence R. Gesquiere, Nancy R. Gee, Kerinne Levy, W. Lance Martin, and C. Sue Carter. “Effects of Affiliative Human-Animal Interaction on Dog Salivary and Plasma Oxytocin and Vasopressin.” Frontiers in Psychology 8, no. SEP (2017): 1–9. https://doi.org/10.3389/fpsyg.2017.01606.
Nagasawa, Miho, Shouhei Mitsui, Shiori En, Nobuyo Ohtani, Mitsuaki Ohta, Yasuo Sakuma, Tatsushi Onaka, Kazutaka Mogi, and Takefumi Kikusui. “Oxytocin-Gaze Positive Loop and the Coevolution of Human-Dog Bonds.” Science 348, no. 6232 (April 17, 2015): 333–36. https://doi.org/10.1126/science.1261022.

373-377: You have not reported anything about COVID-19 response/restrictions.
390-393: Unpack this as an alternative explanation to social bonding.
425-427: unpack this and be more specific.
438-439: please cite the papers you are referring to here.
443-444: I’m not sure why the behavioral coding itself is a limitation, unless you are referring to something about how this was done (e.g. the videos).

---

## Round 0.2 · Minor Revisions

I was very fortunate to have all three original reviewers review your revision. Two of the three are now essentially satisfied with your paper.

However, Reviewer 3 asks for some further qualifications regarding some of your methodological and statistical approaches. I think these suggestions are reasonable and would ask you to undertake another minor revision to address these remaining points. In addition, I agree with the reviewers that there is no need to report both an ANOVA and an ANCOVA. It also needs to be clearer that you focused on an interaction between pre-post scores and experimental condition.

Reviewer 1 ·

Basic reporting

No comment

Experimental design

No comment

Validity of the findings

No comment

Additional comments

The paper is clearly improved as a consequence of the authors' response to the extensive suggestions of the reviewers. The questions and suggestions I previously had, were satisfactorily addressed by the authors. This paper is very interesting and an important addition to increasing knowlegde about the human-cat relationship.

·

Basic reporting

Thank you for your thoughtful response to all my comments.
The manuscript has been well revised and I do not see any major problems.

I think the manuscript has become quite long, as well as the author feels. If you are concerned about it, the section I pointed out about future research (PDS) could be made a bit more compact.

Experimental design

no comment

Validity of the findings

no comment

Additional comments

You have answered my question regarding the limiting of the experimental subjects to reproductive aged women in the Discussion (lines 436-455). I think it might be better to bring this part to the Introduction or Materials & Methods (Participant Screening section).

Reviewer 3 ·

Basic reporting

Many of the textual revisions have improved the clarity of the manuscript. The increased level of detail in the methods is particularly important and helpful. And I would like to reiterate that I do believe this study addresses an important question by exploring the human-cat interaction in a similar way as human-dog interactions have previously been studied.

Experimental design

No comment/see below.

Validity of the findings

Thank you for including figure 1. I think this helps make sense of the data. Please restrict the y-axis to the range of the data so that differences can be seen more clearly.

However, my two most major concerns were not addressed effectively.

1) Lyophilization and validation:

Although the authors are correct that lyophilization of saliva samples is frequently used in the literature, few of the papers they cite perform any validation themselves. Unfortunately, these sorts of methodological details are sometimes overlooked, but they are important, especially given the controversy surrounding measurement of oxytocin.

The current study concentrates samples by 6x, whereas most previous studies only concentrate by 2-4x. Only two of the cited studies use a 6x concentration, and neither of those papers include any analytical validation. I believe the only cited study that performs spike recovery is Carter et al. 2007, which not only used a different concentration, it also used a different kit (Assay Designs) with a different antibody (note also that the Enzo kit began using a new antibody in 2013, as cited in Daughters et al. 2015).

Although lyophilization is commonly used to eliminate matrix interference, it can also magnify matrix interference by concentrating molecules other than the analyte. It is common that there is a “sweet spot” in which assays perform reliably, and it is quite possible that a 6x concentration might be outside of this sweet spot. This is why it is always important to report parallelism (bracketing the concentration of study samples) and spike recovery (at the concentration used for study samples).

Ideally, the exact method—as performed in this lab—should be validated, either here or in another paper (see MacLean et al. 2019 for more on why this is particularly important for oxytocin studies). If that is not possible, it is important to note any deviations in your protocol from previously validated methods and explicitly state that these components of your method have not been validated.

MacLean, Evan L., Steven Ray Wilson, W. Lance Martin, John M. Davis, Hossein P. Nazarloo, and C. Sue Carter. “Challenges for Measuring Oxytocin: The Blind Men and the Elephant?” Psychoneuroendocrinology 107, no. May (2019): 225–31. https://doi.org/10.1016/j.psyneuen.2019.05.018.

2) Difference of logs:
I do not object to the use of a log transform; this is common practice for hormone data, especially when there is right skew (although I will also point out that what matters in determining the necessity of a transformation is the residuals, not the raw distributions). The issue is with taking the difference of already logged concentrations.

log(a) – log(b) = log(a/b), therefore a difference between logged values tells you about the ratio of the two concentrations, not their absolute difference.

Note, that in the two papers cited, MacLean et al. 2017 and Marshall-Pescini et al. 2019, they are primarily modeling the log concentrations with time point as a fixed effect. In cases where they are analysing the association with behavior, however, MacLean et al. 2017 do use the log-transformed percent change, which could be appropriate here as well.

Many people use the absolute or ratio change as the outcome variable in this sort of study. But it is crucial to be clear as to what you are doing. By using a difference of logged values, you are looking at ratio change, and because you are using log10, this can be interpreted as how many orders of magnitude the change between pre- and post- samples is.

Please note also, that the usage of a log-transformation in general should be justified based on it improving the normality of residuals, not simply based on precedent.

Additionally, in looking more closely at the provided data again, I am confused by the raw and adjusted concentrations. The adjusted concentrations appear to multiply the raw values by 6. However, as the samples were concentrated (not diluted) by 6x before assay, I believe the adjusted values should divide the raw concentrations by 6 instead, unless I am misunderstanding the columns. This would affect the reported results as well.

Additional comments

1. Line 83: suggest “intraspecific” before “social behaviors”
2. Line 90: “result from” or “are a result of”?
3. Lines 97-101: from the text alone, this sounds like a single study, but from the citations it looks like two. Please clarify.
4. Line 103: please specify that increased urinary OT was associated with higher levels of mutual gaze.
5. Lines 105-107: it’s great to cite these studies, but I think even without those studies this would be interesting, mostly because cats have also been (semi-)domesticated for thousands of years and are one of the most common household pets. If you agree, I’d suggest saying as much, as I think it provides stronger motivation.
6. Lines 124-132: I disagree that the only solution here is more standardization (although that is likely important on the hormone measurement side of things). Different approaches (e.g. intranasal administration vs. measuring endogenous levels) have different rationales and different pros and cons. This section could be strengthened by considering those, particularly given the abundance of intranasal oxytocin studies and the importance of endogenous oxytocin studies such as this one.

---

## Round 0.3 · accepted · Accept

Thank you for addressing the previous round of comments from reviewers. I believe that your manuscript is now acceptable. However, it is possible that PeerJ requests improvements to Figure 1, which is still hard to read in the latest PDF.

For Table 5, it is odd to report p values to the left of r; typically p values are in the right most column or can be omitted if asterisks are used to denote significance. If possible, reverse the columns during proofs.